# Magnitude of multidrug-resistant and extended-spectrum β-lactamase-producing gram-negative bacteria from tracheal aspirates of intensive care unit patients in Ethiopia

Zenebe Gebreyohannes Berhe[1]*, Shambel Araya Haile[2,3], Gadissa Bedada Hundie[1], Ashenafi Alemu Wami[4], Tesfa Addis[5], Elias Alehegn[1,4], Mahlet Abayneh[6], Shalom Tsegaye Zergaw[6], Natnael Dejene Engida[1], Alganesh Gebreyohanns[1], Firehiwot Workneh[7], Tsedale Woldu Hadgu[1], Yonas Kahasay[1], Kasahun Gorems[1], Rozina Ambachew Geremew[1], Fitsum Girma Teshome[1,5], Tibebe Adinew[1], Daniel Kahase Gebrelibanos[8], Gizachew Taddesse Akalu[1], Semaria Solomon[1]

1 Department of Microbiology, Immunology and Parasitology, St. Paul's Hospital Millennium Medical College, Addis Ababa, Ethiopia, 2 Department of Medical Laboratory Science, College of Health Science Addis Ababa University, Addis Ababa, Ethiopia, 3 Department Molecular and Translational Science, Monash University, Melbourne, Victoria, Australia, 4 Armauer Hansen Research Institute, Addis Ababa, Ethiopia, 5 Ethiopian Public Health Institute, Addis Ababa, Ethiopia, 6 Department of Pediatrics, St. Paul's Hospital Millennium Medical College, Addis Ababa, Ethiopia 7 Addis Continental Institute of Public Health, Addis Ababa, Ethiopia, 8 Wolkite University, Ethiopia

* zenebezoha@gmail.com

## Abstract

### Background

Globally, multidrug-resistant (MDR) and extended-spectrum β-lactamase-producing (ESBL)Gram-negative bacterial pathogens are causing significant public health problems, particularly in intensive care units (ICUs) among patients on mechanical ventilation. The objective of the study was to identify Gram-negative bacterial pathogens from tracheal aspirates, determine the prevalence of MDR, and assess the prevalence of ESBL production among the isolates.

### Methods

A hospital-based cross-sectional study was conducted at St. Paul's Hospital Millennium Medical College (SPHMMC). A total of 181 ICU patients on mechanical ventilation from January to August of 2022 were included. Tracheal aspirates were collected using consecutive sampling techniques, and the samples were inoculated on chocolate, blood, and MacConkey agar plates. Conventional biochemical tests were used to identify Gram-negative bacteria, and antimicrobial susceptibility testing (AST) was carried out via the Kirby–Bauer disc diffusion method. The production of ESBL was confirmed by the combination disc method. The data were entered and analyzed using SPSS version 25 software.

**Data availability statement:** All relevant data are within the article and its supporting information files.

**Funding:** The author(s) received no specific funding for this work.

**Competing interests:** The authors have declared that no competing interests exist.

**Abbreviations:** AOR, Adjusted odds ratio; AST, Antimicrobial susceptibility testing; ATCC, American type culture collection; CLSI, Clinical and laboratory standard institute; COR, Crude odds; ESBL, Extended-spectrum β-lactamase; HAI, Healthcare-associated infection; ICU, Intensive Care Unit; ID, Identification; LRTIs, Lower respiratory tract infections; MDR, Multidrug-resistance; QC, Quality control; SPHMMC, St. Paul's Hospital Millennium Medical College; Spp., Species; SPSS, Statistical Package for Social Sciences; USIN, Unique Survey Identification Number; VAP, ventilator-associated pneumonia, VIF, Variance inflation factor.

## Results

In this study, 181 study participants were enrolled, with an age distribution ranging from 1 year to 81 years and a median and mode of 15 years and 1 year, respectively. The overall prevalence of Gram-negative bacteria was 65.2%. A total of 189 Gram-negative bacteria were found, of which 52 (44.1%) showed a single organism and 66 (55.9%) were multiple organisms. Of the 189 isolates, 69.8% were MDR, while 86.8% were ESBL producers. *Acinetobacter species* (50.8%) and *Klebsiella pneumoniae* (29.6%) were the predominant isolates, with MDRs of 78.1% and 71.4%, respectively, and ESBLs of 100% and 82.1%, respectively.

## Conclusions

A high magnitude of MDR and ESBL was detected. In addition, there was high resistance to cephalosporin-class antibiotics, which is concerning. The MDRs *Acinetobacter spp.* and *K. pneumoniae* were the predominant isolates. Regular monitoring of antimicrobial resistance levels, implementing antimicrobial stewardship, and an effective infection control program should be strengthened.

## Introduction

Ventilator-associated pneumonia (VAP) is a kind of pneumonia that develops more than 48 hours after starting mechanical ventilation and is a common complication of respiratory failure [1–3]. It is one of the most common respiratory infections among critical patients admitted to intensive care units (ICUs) [3,4], and it is mostly linked to long ICU stays and mechanical ventilation support [5]. According to the World Health Organization (WHO), ICUs have some of the highest rates of healthcare-associated infections (HAIs), owing primarily to the use of indwelling medical equipment such as endotracheal tubes, which raises the risk of infection [6]. Furthermore, ICU conditions, including frequent patient contact by healthcare providers, environmental reservoirs, inadequate infection prevention, immunity status, and high antibiotic usage, play an important role in the growth and spread of multidrug-resistant microorganisms in healthcare settings [7]. The incidence of HAI in ICUs is around 2–5 times higher than in the general hospital population [8]. Lower respiratory tract infections (LRTIs) are the most common diseases encountered, with approximately 10–25% of ICU patients developing LRTIs and a death incidence ranging from 22% to 71% [9].

Antimicrobial resistance (AMR) has emerged as a serious global public health problem, especially in ICUs [10]. The rise of multidrug-resistant (MDR) Gram-negative bacteria is especially alarming, as these organisms are resistant to a wide range of drugs, creating significant treatment challenges [7,8]. Many Gram-negative bacteria produce extended-spectrum beta-lactamases (ESBL), which enable them to resist broad-spectrum beta-lactam antibiotics. These ESBL-producing bacteria are major contributors to HAIs, resulting in longer hospital stays, greater morbidity and mortality rates, and higher healthcare costs [11,12]. Beta-lactamases are the main mechanism of bacterial resistance to beta-lactam antibiotics, making infections caused by MDR- and ESBL-producing bacteria extremely challenging to treat [13].

HAIs are avoidable infections that arise in healthcare settings; ICU patients are 5–10 times more likely to develop HAIs than patients in other hospital units [14,15]. VAP is responsible for 90% of HAIs in ICUs and often develops 48 hours or more after artificial breathing is started via an endotracheal tube [16,17]. Gram-negative bacteria are responsible for 45%–70% of LRTIs in ICUs, with many of these pathogens being MDR [15].

The rise of antibiotic resistance has significantly complicated the treatment of infections caused by Gram-negative bacteria [18–20]. Worldwide, resistance to fluoroquinolones and the prevalence of ESBL-producing Enterobacterales are increasing [21]. ESBL-producing Gram-negative bacteria are currently the leading cause of HAIs, and resistance to third- and fourth-generation cephalosporins caused by acquired ESBLs is an increasing issue [3,10,22].

Bacteria in healthcare facilities are growing more resistant to conventional antibiotics, particularly in intensive care units [23]. The rapid spread of MDR and ESBL-producing Gram-negative bacteria in ICUs has posed a serious challenge to infection control and patient care, becoming a significant challenge for infection management and patient care [24].

AMR infections are common in low- and middle-income countries like Ethiopia [25]. Globally, consumption of anti-biotics showed an increase of 65% in studies conducted between 2000 and 2015 years driven by high utilization in low- and middle-income countries [26] especially high inappropriate utilization of antibiotics in health facilities was reported from African countries [27–30]. Antibiotic dispensing without a prescription remains an important factor among sub-Saharan countries, ranging up to 89.2% in Eritrea, Ethiopia (up to 87.9%), Nigeria (up to 86.5%), Tanzania (up to 92.3%), and Zambia (up to 100%) [31]. This is associated with a lack of surveillance systems, poor infection prevention and control, non-adherence to treatment guidelines, loose rules and regulations and limited microbiology infrastructures [32,33]. The Ethiopian national AMR surveillance plan was launched in April 2017; however, it faces challenges such as limitations of data capture, gaps of supply including a lack of rapid diagnostic devices in microbiology laboratories, communication barriers, and prolonged turnaround time of microbiological tests, which has limited the timely initiation of pathogen-specific therapy [34].

This study was conducted to determine the prevalence of MDR, and ESBL-producing Gram-negative bacteria among ICU patients using tracheal aspirate samples. The study findings may help in the development of hospital-specific antibi-otic usage recommendations as well as in providing initial information for future studies.

## Materials and methods

### Study design, period and setting

A hospital-based cross-sectional study was conducted from January to August of 2022 at SPHMMC, Addis Ababa, Ethi-opia. The SPHMMC is a government-affiliated hospital that serves as a teaching center for healthcare professionals and offers tertiary referral hospital care to patients from different parts of the country. The laboratory work was conducted in the Microbiology, Immunology, and Parasitology Departments.

### Study population

All patients diagnosed with LRTIs were admitted to the SPHMMC's ICU units and received mechanical ventilation support.

### Inclusion and exclusion criteria

This study included patients hospitalized in the ICU with LRTIs who had received mechanical ventilation for 48 hours or more, under the usual diagnosis of VAP, which typically develops after 48 hours of mechanical ventilation [35]. Patients who were under ventilation for less than 48 hours, who had tuberculosis, or who were receiving anti-TB empirical medi-cation during the study period were excluded because they were going to be transferred to other TB treatment institutions and we excluded tuberculosis patients due to the distinct microbiological and clinical characteristics of Mycobacterium tuberculosis, which require specialized diagnostic and management approaches outside the scope of this study.

## Data collection

A structured questionnaire was used to collect information about socio-demographic characteristics and clinical data (age, sex, empirical treatment history, history of asthmatic conditions, residency area, duration of admission, duration of mechanical ventilation, previous TB cases, use of invasive medical devices, duration of admission, type of antibiotic given, etc.) after informed assent was signed by the study participant's guardian, and other necessary information was collected from their medical card.

## Sample collection techniques

The tracheal aspirate is chosen due to its critical role in diagnosing VAP, offering a direct and reliable indication of lower respiratory tract infections, addressing clinically significant concerns in our clinical setting, and aligning with the study's focus on MDR in VAP-related pathogens. Experienced nurses working in the ICU units collected tracheal aspirate samples from patients suspected of having LRTIs using aseptic techniques, including sterile gloves, catheters, and collection containers. The catheter was carefully inserted without contact with the upper airway into the endotracheal tube until resistance was encountered (at the level of the carina in the trachea), and the tube was retracted approximately 2 cm. After gentle aspiration without the addition of saline, the aspirate was transferred to leak-proof and sterile tubes with 15 ml of graduated conical Falcon tubes, and the samples were subsequently sent under optimal conditions to the microbiology laboratory for processing within less than one hour to prevent bacterial overgrowth or degradation.

## Inoculation and identification of bacteria

The standard of acceptance for obtaining tracheal specimens from adult and pediatric patients was based on Gram staining. For adults, we used at least 10 squamous epithelial cells per low-power field and no organism [36], and the absence of organisms on Gram staining was a criterion for rejecting pediatric patients [37].

The accepted tracheal aspirates were inoculated immediately on blood agar, chocolate agar, and MacConkey agar plates and incubated at 37°C. Blood and chocolate agar plates were incubated in 5% $CO_2$ jars; on the other hand, MacConkey was incubated aerobically. Growth was observed after 24 and 48 hours of incubation. The colonies on the positive plates were preliminarily characterized by colony characterization and a Gram-stain reaction. The isolated organisms were processed with conventional biochemical tests, including oxidase, triple sugar iron agar, citrate utilization, urea, mannitol, decarboxylation, hydrogen sulfide, indole, and motility tests, for identification.

## Antimicrobial susceptibility testing (AST)

The ASTs of each isolate were tested on Mueller–Hinton agar (Liofilchem, Italy) using the standard Kirby–Bauer's disc diffusion method, which was based on the Clinical and Laboratory Standard Institute (CLSI) 2022 M100 guidelines [38]. Before streaking to the agar plate, the inoculum was adjusted to a turbidity equivalent to a 0.5 McFarland standard by placing the tubes in the DEN-1B densitometer. After the appropriate antibiotic discs were applied and incubated for 15 minutes, the plates were inverted and incubated at 37°C. After 18–24 hours of incubation, each plate was examined, and the diameter of the zone of inhibition was measured in millimeters and interpreted as per CLSI 2022 breakpoints [38]. After the diameter of the inhibition zone was measured, the organisms were reported as susceptible, intermediate, or resistant. MDR bacteria were described as being resistant to at least one antimicrobial agent in three or more antimicrobial categories [16,18,28,39]. In this study, the antimicrobial agents tested for AST were amikacin (Ak-30 μg), ceftriaxone (Ctr-30 μg), ceftazidime (Caz-30 μg), cefepime (Cep-30 μg), ciprofloxacin (Cip-5 μg), sulfamethoxazole-trimethoprim (SXT 1.25/23.75 μg), meropenem (Mero-10 μg) and imipenem (Imp-10 μg). These antimicrobial agents were chosen based on CLSI criteria and their relevance in controlling MDR bacteria. Carbapenems, in particular, were chosen to treat critically ill patients in intensive care units who developed VAP as a result of ESBL and carbapenemase-producing infections, which are common in our region and widely prescribed in our settings as a last-resort of treatment. All the antibiotic discs used had been Oxoid, (UK) products.

                                                                                       

### Screening and confirmation of ESBL-producing isolates

The screening and confirmation of ESBL-producing Gram-negative bacteria were performed by using the antibiotic disc diffusion method on MHA plates with the bacterial suspension adjusted to 0.5 McFarland and incubated overnight at 37°C. Isolates that exhibited an inhibition zone size ≤22 mm for ceftazidime and/or ≤27 mm for cefotaxime (30 µg) were considered to be ESBL producers. These isolates were further confirmed by a combination disc method. Ceftazidime (30 µg) and ceftazidime were combined with clavulanic acid (30 µg + 10 µg), and cefotaxime and cefotaxime were combined with clavulanic acid (30 µg + 10 µg) discs. ESBL-producers were identified by a zone diameter increase of ≥5 mm around the disc with the antibiotic in combination with clavulanic acid [38].

### Quality assurance

All culture media were prepared according to the manufacturer's instructions; the sterility of the prepared media was verified by overnight incubation at 37°C. The performance of the culture media, including biochemical tests and antibiotic discs, was checked using American Type Culture Collection (ATCC) control strains. The ATCC strains used were S. aureus (ATCC 25923), E. coli (ATCC 25922), P. aeruginosa (ATCC 27853), *P. mirabilis (ATCC 35659)*, *K. pneumoniae (ATCC 700603), S. pneumoniae (ATCC 49619), and E. faecalis (ATCC 29212)*. For the ESBL confirmatory test, *K. pneumoniae* ATCC 700603 (ESBL producer) and *E. coli* ATCC 25922 (ESBL negative) control strains were used.

Each sample was labeled with a unique survey identification number (USIN) as indicated in the questionnaire. The questionnaire was first prepared in English and subsequently translated into Amharic and returned to English to check for consistency. Additionally, the questionnaire was pretested and checked for completeness before and during the data collection.

### Data analysis

All data have been entered, checked, cleaned, and analyzed using the Statistical Package for Social Sciences (SPSS) version 25 software. We carefully worked through source documents such as laboratory results and patient files to assess data completeness as well as missing or incorrectly entered information, if any. We made efforts to go back to the original records and consult with site data collectors to retrieve missed data. Descriptive statistics of frequency and percentage were used to present the prevalence of the target bacteria and for variables of socio-demographic distribution. The 95% confidence interval was estimated. We assessed the relationship between the dependent and independent variables using bivariate analysis. The variables included in the logistic regression model were selected based on clinical relevance and statistical significance in univariate analyses with a p-value less than 0.25 and were candidates for multivariate logistic regression analysis. Before interpreting the findings, we calculated the variance inflation factor (VIF) and tolerance to verify the impact of multicollinearity among the independent variables. In multivariate analysis, a P-value less than 0.05 was considered statistically significant.

### Ethics approval and informed consent

Ethical approval was obtained from the Institutional Review Board (IRB) of the SPHMMC (PM23/279). The purpose of the study was clearly explained to each study participant's guardian, and written consent was obtained. The study participants' privacy was strictly maintained, and the study was performed as per the Helsinki Declaration [40].

## Results

### Socio-demographic characteristics of the study participants

A total of 181 patients with LRTIs on mechanical ventilation were enrolled in this study. Nearly 50% (n = 90) were pediatric ICU patients. The median and mode ages were 15 years and 1 year, respectively, with the age distribution ranging from 1 year to 81 years. Almost half of the study participants were less than 16 years old (**Table 1**).

### Frequency, multidrug-resistance, and ESBL profile of the Gram-negative isolates

The most frequently found isolates in this study were *Acinetobacter spp.* and *K. pneumonia,* with a prevalence of 96 (50.8%) and 56 (29.6%), respectively. Overall, 130 (68.8%) bacterial isolates were shown to be MDRs. MDR strains were found in 78.1% of *Acinetobacter spp.*, followed by 71.4% of *K. pneumoniae*. While 86.8% of the isolated Gram-negative bacteria were ESBL-producers. Among these, a high prevalence of ESBL-producers was observed among the isolates of *Acinetobacter spp.* (100%) and *K. pneumoniae (*82.1%) (Table 2*).

### The pattern of antibiotic resistance for gram-negative isolates

A high resistance level was observed for *Acinetobacter spp.* in all drug classes, especially in the cephalosporin class ranging from 86.5% to 95.8%. Most of the isolates exhibited high resistance to cephalosporin and sulfamethoxazole-trimethoprim (Table 3).

### Factors associated with the level of lower respiratory tract infection

In bivariate logistic regression analysis, ICU units (P = 0.023), length of hospital stay in ICU (P = 0.009), residence areas (P = 0.049), linked from departments (P = 0.049), and length of hospital stay (P = 0.005) were the candidate variables for multivariate logistic regression analysis. In multivariate analysis, it was revealed that patients admitted to the PICU were nearly three times more likely to be diagnosed with LRTIs than those admitted to the AICU (AOR = 2.982, 95% CI = 1.98–2.803, P = 0.001). Additionally, prolonged hospitalization of 15 days or more was strongly associated with an increased

**Table 1. Socio-demographic characteristics of the study participants at the ICU of SPHMMC (December 2021 – July 2022).**

| Variables | Categories | Frequency (%) |
|---|---|---|
| Sex | Male | 94(51.9) |
| | Female | 87(48.1) |
| Age(years) | ≤15 | 91(50.3) |
| | 16-30 | 27(14.9) |
| | 31-45 | 25(13.8) |
| | 46-60 | 17(9.4) |
| | ≥61 | 21(11.6) |
| Residence areas | Urban | 143(79) |
| | Rural | 38(21) |
| ICU units | AICU | 91(50.3) |
| | PICU | 90(49.7) |
| Sources of ICU admission | Surgery department | 25(13.8) |
| | Pediatrics emergency | 79(43.6) |
| | Internal medicine | 13(7.2) |
| | Adult emergency | 64(35.4) |
| Length of ICU admission | <7 day | 128(70.7) |
| | 7-15 day | 29(16) |
| | >15 day | 24(13.3) |
| Empirical treatment | Yes | 179(98.9) |
| | No | 2(1.1) |

Keys: ICU-intensive care unit, AICU-Adult ICU, PICU-Pediatrics ICU.

**Table 2. Distribution of MDR and ESBL-producing gram-negative bacteria in ICU units SPHMMC (December 2021- July 2022), N = 189.**

| Bacteria Species | Total Isolates (n, %) | AICU (n, %) | PICU (n, %) | MDR (AICU, n, %) | MDR (PICU, n, %) | Total MDR (n, %) | ESBL (AICU, n, %) | ESBL (PICU, n, %) | Total ESBL (n, %) |
|---|---|---|---|---|---|---|---|---|---|
| *Acinetobacter spp.* | 96(50.8%) | 38(39.6%) | 58(60.4%) | 34(89.5%) | 41(70.7%) | 75(78.1%) | 38(100%) | 58(100%) | 96 (100%) |
| *E. coli* | 22(11.6%) | 11 (50%) | 11 (50%) | 3(27.3%) | 9(81.8%) | 12(54.5%) | 5(45.5%) | 9(81.8%) | 14(63.6%) |
| *K.pneumoniae* | 56(29.6%) | 20(35.7%) | 36(64.3%) | 16 (80%) | 24(66.7%) | 40(71.4%) | 16 (80%) | 30(83.3%) | 46(82.1%) |
| *K. ozaenae* | 3 (1.6%) | 1 (33.3%) | 2 (66.7%) | 1 (100%) | 1 (50%) | 2 (66.7%) | 1 (100%) | 2 (100%) | 3 (100%) |
| *Pseudomonas spp.* | 8 (4.2%) | 3 (37.5%) | 5 (62.5%) | 0 (0%) | 0(0%) | 0 (0%) | 0(0%) | 5 (100%) | 5 (62.5%) |
| *E. cloacae* | 1 (0.5%) | 1 (100%) | 0(0%) | 1 (100%) | 0 (0%) | 1 (100%) | 0 (0%) | 0 (0%) | 0 (0%) |
| *Proteus spp.* | 2 (1.1%) | 2 (100%) | 0 (0%) | 1 (50%) | 0 (0%) | 1 (50%) | 0 (0%) | 0 (0%) | 0 (0%) |
| *Serratia spp.* | 1 (0.5%) | 1 (100%) | 0 (0%) | 1 (100%) | 0(0%) | 1 (100%) | 0 (0%) | 0 (0%) | 0 (0%) |
| Total | 189(100%) | 77(40.7%) | 112(59.3%) | 57 (74%) | 75 (67%) | 132(69.8%) | 60(77.9%) | 104(92.9%) | 164(86.8%) |

Keys: MDR: Multidrug-resistant, ESBL: Extended-spectrum beta-lactamase producers, AICU: Adult Intensive Care Unit, PICU: Pediatric Intensive Care Unit, MDR (n, %): Number and percentage of multidrug-resistant isolates, ESBL (n, %): Number and percentage of extended-spectrum beta-lactamase-producing isolates.

**Table 3. Antibiotic resistance patterns of bacterial isolates from tracheal aspirates at the ICU of SPHMMC (December 2021- July 2022), N = 189.**

| Type of bacterial isolates (n = total number) | Drug classes and number of isolates (% resistance) | | | | | | | |
|---|---|---|---|---|---|---|---|---|
| | Cephalosporins | | | Fluoroquinolone | Aminoglycoside | Carbapenem | | DHFR inhibitor |
| | CAZ (30 µg) | CRO (30 µg) | CEP (30 µg) | CIP (5 µg) | AK (30 µg) | MRP (10 µg) | IMP (10 µg) | SXT (1.25/23.75 µg) |
| *Acinetobacter spp.* (n = 96) | 89 (92.7) | 92(95.8) | 83(86.5) | 70 (72.9) | 39 (40.6) | 52(54.2) | 47(49) | 83 (86.5) |
| *E. coli* (n = 22) | 15(68.2) | 19(86.4) | 14(63.6) | 10 (45.5) | 4 (18.2) | 3(13.6) | 4(18.2) | 20 (90.9) |
| *K. pneumoniae* (n = 56) | 37(66.1) | 52(92.9) | 50(89.3) | 32(57.1) | 5(8.9) | 14(25) | 16(28.6) | 44(78.6) |
| *K. ozaenae* (n = 3) | 2 (66.7) | 3 (100) | 3 (100) | 1(33.3) | 1(33.3) | 2(66.7) | 2(66.7) | 3(100) |
| *Proteus spp.* (n = 2) | 1(50) | 2(100) | 0(0) | 1(50) | 0(0) | 0(0) | 0(0) | 1(50) |
| *Pseudomonas spp.* (n = 8) | 4 (50) | ND | 2 (25) | 2 (25) | 0 (0) | 0 (0) | 0 (0) | ND |
| *Serratia spp.* (n = 1) | 1(100) | 1(100) | 1(100) | 1(100) | 0 (0) | 0 (0) | 0 (0) | 1(100) |
| *E. cloacae* (n = 1) | 1(100) | 1(100) | 1(100) | 1(100) | 1(100) | 1(100) | 1(100) | 1(100) |

Keys: ND, not done; 0, 100% sensitive; Caz, ceftazidime. CRO: ceftriaxone, CEP: cefepime, CIP: ciprofloxacin, AK: amikacin, Mero: meropenem, Imp: imipenem, SXT: sulfamethoxazole-trimethoprim/DHFR: dihydrofolate reductase/Folate pathway inhibitors.

risk of developing LRTIs (AOR = 0.171, 95% CI = 0.029–0.988, P = 0.048). The detailed information on the bivariate and multivariate logistic regression analyses is summarized in (Table 4).

## Discussion

In clinical settings, microorganisms are becoming significantly more resistant to conventional antibiotics, leading to MDR. Critically ill patients receiving mechanical ventilation in ICUs will progressively develop VAP infection [23]. The rapid spread of MDRs and ESBL-producing Gram-negative bacteria infections in ICUs has become a major challenge [24]. Therefore, this study aimed to determine the magnitude of MDRs and ESBL-producing Gram-negative bacteria among ICU patients with LRTIs at SPHMMC. The overall magnitude of the Gram-negative bacteria in this study was 118 (65.2%), with a high percentage of MDR and ESBL-producing isolates identified.

**Table 4. Bivariate and multivariate logistic regression analyses of factors associated with LRTI at the ICU of SPHMMC (December 2021- July 2022).**

| Variables | Categories | Bacterial growth | | COR (95%CI) | P-value | AOR (95%CI) | P-value |
|---|---|---|---|---|---|---|---|
| | | Yes | No | | | | |
| Sex | Male | 64 | 30 | 0.767(0.416-1.416 | 0.396 | – | – |
| | Female | 54 | 33 | 1* | | – | – |
| Age (years) | ≤15 | 67 | 24 | 0.716(0.258-1.987) | 0.522 | – | – |
| | 16-30 | 14 | 13 | 1.857 (0.571-6.046) | 0.304 | – | – |
| | 31-45 | 17 | 8 | 0.941(0.273-3.241) | 0.923 | – | – |
| | 46-60 | 6 | 11 | 3.667(0.954-14.092) | 0.59 | – | – |
| | ≥61 | 14 | 7 | 1* | | – | – |
| ICU units | AICU | 52 | 39 | 1* | | 1* | 0.001 |
| | PICU | 66 | 24 | 0.485(0.260-0.906) | **0.023** | 1.982(1.98-2.803) | |
| Length of stay in ICU | <7 days | 78 | 50 | 1* | | | |
| | 7-15 days | 17 | 12 | 14.744(1.930-112.637 | **0.009** | 0.175(0.019-1.657) | 0.129 |
| | >15 days | 23 | 1 | 16.235(1.921-137.185) | **0.002** | 0.122(0.012-1.211) | 0.072 |
| Residence areas | Urban | 88 | 55 | 2.344(1.002-5.481) | **0.049** | 0.621(0.134-7.28) | 0.07 |
| | Rural | 30 | 8 | 1* | | | |
| Sources of ICU admission | Surgery department | 17 | 8 | 1* | | | |
| | Adult emergency | 36 | 28 | 1.219(0.460-3.228) | 0.690 | | |
| | Internal medicine | 8 | 5 | 2.015(1.004-4.046) | **0.049** | 1.810(0.238-13.761) | 0.566 |
| | Pediatrics emergency | 57 | 8 | 1.619(0.478-5.490) | 0.439 | | |
| Length of hospital stay | <7 days | 56 | 39 | 1* | | | |
| | 7-15 days | 37 | 22 | 8.705(1.948-38.904) | **0.005** | 0.227(0.041-1.248) | 0.088 |
| | >15 days | 25 | 2 | 7.432(1.603-34.458) | 0.010 | 0.171(0.029-0.988) | 0.048 |

Keys: AOR-Adjusted odds ratio, COR-Crude odds ratio, 1*- reference for the categories, bold p-values are statistically significant.

In the present study, the prevalence of Gram-negative bacteria from tracheal aspirates was comparable to that in previously reported studies done in Pakistan, India, and Nepal respectively: 68.4% by Ahmad H, *et al.* [23], 65.2% by Agarwal S, *et al.* [41], 67% by Swati A, *et al.* [42], 68.08% by Shrestha R, *et al.* [6].

In contrast, the present study revealed more significant bacterial infections than did the previous studies by Ahmed NH, *et al.* [43], Jakribettu RP, *et al.* [44], Jethwani U, *et al.* [45] and Syal K, *et al.* [46], with findings of 48.04%, 44.02%, 34.5%, and 56.3%, respectively. These variations in infection rates may be due to factors of infection control practices, hospital crowding, study design, patient type, and clinical conditions. During the study, several contributing factors were observed, including high patient volume, shared gowns among attendants, crowded rooms with closely spaced beds, and compromised disinfection of ICU equipment's, including mechanical ventilators.

However, the prevalence of Gram-negative bacteria from tracheal aspirates in our result was less compared to previous similar studies conducted by Malik MI, *et al.* [47], Banerjee A, *et al.* [48], Sattar FAA, *et al.* [49], Vadivoo NS, *et al.* [50], Koirala P, *et al.* [51] and Rashid O, *et al.* [52], Gupta P, *et al.* [53] and Batool A, *et al.* [54], their findings ranged from 73% to 92%. Furthermore, the prevalence of Gram-negative bacteria in our study was lower than that reported by Prajapati BK, *et al.* [55], Rathod VS, *et al.* [56], Samal N, *et al.* [57], Khanal S, *et al.* [13] and Khoshfetrat MK, *et al.* [58] which were 97%, 80%, 85.7%, 92.2%, and 82.7%, respectively. This discrepancy may result from variations in the study's duration, study design, infection control strategies, and geographic setting.

In the present study, *Acinetobacter spp.* (50.8%), followed by *K. pneumoniae* (29.6%), were the two most common Gram-negative bacteria isolates. This finding is in agreement with previously similar reported studies of Shrestha R, *et al.* [6], Khanal S, *et al.* [13], Swati A, *et al.* [42], Jethwani U, *et al.* [45], Batool A, *et al.* [54] and Shankare, *et al.* [59]. However, other similar studies have identified *Klebsiella spp.* and *P. aeruginosa* as the predominant isolates, as reported by Jakribettu RP, *et al.* [33], Malik MI, *et al.* [36], Gupta P, *et al.* [42], Rathod VS, *et al.* [45], Brusselaers N, *et al.* [24] and Shalini S, *et al.* [60]. Moreover, Khoshfetrat M, *et al.* [58] and Goel V, *et al.* [1] *specifically* reported that *A.baumannii and P. aeruginosa as* the most prevalent pathogens. Similarly, Prajapati BK, *et al.* [55] and Chandra D, *et al.* [61] found *Klebsiella spp.* and *Acinetobacter spp.* to be the most common organisms. On the other hand, Swati A, *et al.* and Shankare, *et al.* [42,59] reported *P. aeruginosa, Acinetobacter spp.*, and *K. pneumoniae* as the most isolated organisms. Furthermore, Patil DT, *et al.* [62] reported *P. aeruginosa* and *K. pneumoniae as* the most commonly isolated organisms. In contrast, Samieeifard S, *et al.* [63] reported *E. coli* and *Enterobacter spp.* as the most prevalent isolates. These variations could be related to a variety of factors, including the type of commensal carriage in the patient population, the number of patients admitted to a single room, and the duration of the ICU stay. Collectively, these data emphasize the dynamic and complex nature of bacterial epidemiology in many environments.

In the present study, a significant number (68.8%) of the Gram-negative bacteria were found to be MDR. Notably, the resistance levels of *Acinetobacter spp.* to third- and fourth-generation cephalosporins were astonishingly high and comparable to those reported by Golli AL, *et al.* [10], Samal N, *et al.* [57], and Nwadike VU, *et al.* [64]. However, the resistance level of this strain to carbapenems was greater than that reported in the above-mentioned studies which were (18.2%) and (35.7%) respectively. The prevalence of MDR found in this study might be associated with the high frequency of mis prescription and inappropriate use of antibiotics; these practices possibly exert selective pressure, resulting in the rise of MDR- strains.

A different pattern of AMR was observed in previous studies conducted by various scholars. In addition, Ahmed NH, *et al.* [43], Syal K, *et al.* [46], and Ahmad H, *et al.* [23] reported that *Acinetobacter spp.* was entirely resistant to cephalosporin and ciprofloxacin. Similarly, *Acinetobacter spp.* strains resistant to meropenem (89%), amikacin (95%), and ciprofloxacin (90%) were reported by Shankare, *et al.* [59], and 95.1% resistance to meropenem was reported by Khoshfetrat M, *et al.* [58]. According to the findings of Saha MR, *et al.* [65], *Acinetobacter spp.* and *Klebsiella spp.* were exhibited as highly resistant. Among those strains, 90–100% were resistant to third-generation cephalosporins and aminoglycosides, while 95–97% were resistant to third-generation cephalosporins and ciprofloxacin; these results are quite similar to the results we obtained. According to the findings of Ahmed NH, *et al.* [43], Gupta P, *et al.* [53], and Ahmad H, *et al.* [23], *Klebsiella spp.* was entirely resistant to cephalosporins and ciprofloxacin. Furthermore, Gram-negative bacteria that were completely resistant to ceftazidime and sulfamethoxazole-trimethoprim were reported by Jethwani U, *et al.* [45]. Our study showed that the isolates were relatively susceptible to carbapenem and amikacin, which is in line with the finding of Jakribettu RP, *et al.* [44]. In our investigation, *Klebsiella spp.* exhibited less resistance to carbapenem, cephalosporins, amikacin ciprofloxacin, and sulfamethoxazole-trimethoprim, as reported by Shankare, *et al.* [59]. According to Rathod VS, *et al.* [56] reports *Klebsiella spp.* and *E. coli* showed no resistance to carbapenem drugs. While our findings revealed a higher resistance level to the carbapenems. On the other hand, we found that *Pseudomonas spp.* was 100% sensitive to imipenem, which is in concordance with the findings of studies conducted by Vadivoo NS, *et al.* [50] and Rathod VS, *et al.* [56].

In our study, we analyzed the percentage of MDR strains among the isolates by taking into consideration resistance to at least three different antibiotic groups: aminoglycosides (amikacin), cephalosporins (ceftazidime, cefepime and ceftriaxone), carbapenems (meropenem and imipenem), sulfamethoxazole-trimethoprim, and fluoroquinolones (ciprofloxacin). The overall MDR level in this study was similar to that reported in a previous study (70.1%) by Khanal S, *et al.* [13]. According to our findings, the MDR levels of *Acinetobacter spp.* and *K. pneumoniae* were 78.1% and 71.4%, respectively, which is equivalent to the findings of studies revealing MDR strains of *Acinetobacter spp.* (85.4%) and *K. pneumoniae* (73.8%) reported by Khanal S, *et al.* [13] and Golli AL, *et al.* [10]· in which the MDR levels were 85.88% and 70.04%,

respectively, for *Acinetobacter spp.* and *K. pneumoniae*. In the present study, we found higher MDR levels in *Acinetobacter spp.* and *K. pneumoniae* than in similar studies, with reported values of 56% and 53%, respectively [50]. However, our finding is lower than that of Goel V, *et al.* [1], who reported 100% MDR of *Acinetobacter spp.* This might be associated with the unnecessary misuse of more generations of antibiotics, empirical treatment, length of stay in the ICU, age, and comorbidities of the study participants.

ESBLs are a group of plasmid-mediated, diverse, complex, and rapidly evolving enzymes that are posing major therapeutic challenges in hospitalized patients seeking treatment. ESBLs are enzymes that act by inhibiting the action of beta-lactam antibiotics and confer resistance to penicillin (excluding temocillin), first, second, and third-generation cephalosporins, and aztreonam (but not cephalosporins or carbapenems), while being inhibited by β-lactamase inhibitors such as clavulanic acid [18,66]. In the present study, we found that the majority of the isolates (86.8%) were ESBL producers. Interestingly, *Acinetobacter spp.* (100%), *Klebsiella spp.* (82.1%), and *E. coli* (63.6%) were the top three ESBL- producers. This result is greater than that of previously reported studies of Joseph NM, *et al.* [2], Swati A, *et al.* [42], Jethwani U, *et al.* [45], and Samal N, *et al.* [46]. The reasons could be related to the antimicrobial used for empirical treatment, high consumption and selective pressure toward the cephalosporin drug class, insufficient empirical use of antimicrobial agents in primary health care facilities, or prolonged hospitalization with a variety of antibiotics. Our findings indicate a high prevalence of MDR and ESBL, with nearly all participants receiving empirical treatment. Although we did not study the trend of AMR, recent surveillance of over 4,000 health centers in Ethiopia has revealed a rising trend in antimicrobial resistance. The antimicrobial resistance level in the causative agent of typhoid fever increased from 10% in 2006 to 87% in 2019 and similarly, *E. coli* strains showed increased resistance to stronger antimicrobial agents [67]. A retrospective analysis from Ethiopia on clinical samples presented a rising trend in MDR infections. The resistance of *Acinetobacter spp.* to ceftriaxone increased from 66.7% to 82.2%, and to ciprofloxacin from 58.5% to 66.7% [68]. *K. pneumoniae* also had a statistically increased resistance to ciprofloxacin and carbapenems, presenting an alarming trend in antimicrobial resistance in the country [69]. These findings suggest that an increasing trend of AMR, particularly in resource-limited settings, is contributing to the breakdown of a last-resort of antibiotics.

## Limitations of the study

Owing to limited resources, assessments of several common pathogens of LRTI, such as gram-positive cocci, *Mycoplasma spp.*, *Legionella spp.*, and fungal etiology, have not been performed. We focused on Gram-negative bacteria due to their high prevalence and significant role in MDR in our setting. While this limits the study's relevance to other pathogen groups, it does allow for an extended examination of Gram-negative infections and their resistance patterns. Furthermore, our research method captures associations at a single point in time, which could underestimate the temporal correlations between variables. Another limitation is that not all antibiotics for Gram-negative bacteria were tested due to the lack of antibiotic discs. However, we selected all the antibiotics used as hospital treatment protocol recommendations.

## Conclusions and recommendations

In this study, the overall prevalence of Gram-negative bacteria was 65.2%. We conclude that the MDRs of *Acinetobacter spp.* and *Klebsiella spp.* were the most common etiological agents isolated from tracheal aspirates with a high resistance to cephalosporin-class antibiotics. Most of the isolates were resistant to multiple classes of antimicrobial agents, with alarming MDRs and ESBL prevalence. Regular monitoring of antimicrobial resistance levels, implementing antimicrobial stewardship, and an effective infection control program should be strengthened.

## Supporting information

**S1 File. Sociodemographic, isolates and antimicrobial susceptibility patterns.**
(XLSX)

## Acknowledgments

The authors would like to extend their sincere gratitude to all of the staff members of the St. Paul's Hospital Millennium Medical College ICU and microbiology departments. We are also grateful to acknowledge the study participants for their willingness and involvement in this study.

## Author contributions

**Conceptualization:** Zenebe Gebreyohannes Berhe, Gizachew Taddesse Akalu, Semaria Solomon.

**Data curation:** Zenebe Gebreyohannes Berhe, Shambel Araya Haile, Fitsum Girma Teshome, Daniel Kahase Gebrelibanos, Tibebe Adinew.

**Formal analysis:** Zenebe Gebreyohannes Berhe, Shambel Araya Haile, Tibebe Adinew.

**Investigation:** Zenebe Gebreyohannes Berhe, Alganesh Gebreyohanns, Rozina Ambachew Geremew, Tsedale Woldu Hadgu, Yonas Kahasay, Natnael Dejene Engida, Shalom Tsegaye zergaw, Mahlet Abayneh.

**Methodology:** Zenebe Gebreyohannes Berhe, Semaria Solomon.

**Project administration:** Zenebe Gebreyohannes Berhe, Firehiwot Workneh.

**Resources:** Ashenafi Alemu Wami, Tesfa Addis, Rozina Ambachew Geremew.

**Software:** Zenebe Gebreyohannes Berhe, Elias Alehegn Workineh, Kasahun Gorems, Semaria Solomon.

**Supervision:** Mahlet Abayneh, Gizachew Taddesse Akalu, Semaria Solomon.

**Writing – original draft:** Zenebe Gebreyohannes Berhe.

**Writing – review & editing:** Zenebe Gebreyohannes Berhe, Shambel Araya Haile, Gadissa Bedada Hundie, Tesfa Addis, Elias Alehegn Workineh, Firehiwot Workneh, Gizachew Taddesse Akalu, Semaria Solomon.

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
