## [Decision Letter · Decision Letter 0]

25 Oct 2024

Dear Dr. Gebreyohannes,

plosone@plos.org. A rebuttal letter that responds to each point raised by the academic editor and reviewer(s). You should upload this letter as a separate file labeled 'Response to Reviewers'.A marked-up copy of your manuscript that highlights changes made to the original version. You should upload this as a separate file labeled 'Revised Manuscript with Track Changes'.An unmarked version of your revised paper without tracked changes. You should upload this as a separate file labeled 'Manuscript'.

We look forward to receiving your revised manuscript.

Kind regards,

Helen Howard

Staff Editor

PLOS ONE

Journal Requirements:

**Additional Editor Comments:**

The manuscript has been evaluated by two reviewers, and their comments are available below.

The reviewers have raised a number of major concerns. They feel the manuscript requires a stronger rationale, improved writing clarity, and they request improvements to the reporting of methodological aspects of the study.

Could you please carefully revise the manuscript to address all comments raised?

Reviewers' comments:

Reviewer's Responses to Questions

**Comments to the Author**

1. Is the manuscript technically sound, and do the data support the conclusions?

Reviewer #1: Yes

Reviewer #2: Yes

2. Has the statistical analysis been performed appropriately and rigorously?

Reviewer #1: I Don't Know

Reviewer #2: Yes

3. Have the authors made all data underlying the findings in their manuscript fully available?

Reviewer #1: Yes

Reviewer #2: No

4. Is the manuscript presented in an intelligible fashion and written in standard English?

Reviewer #1: No

Reviewer #2: Yes

Reviewer #1: Although the topic is important, I find that it lacks novelty because there are lots of studies that have investigated the prevalence of MDR and ESBL in Gram negative bacteria. Authors need to show how is their work different and what is the important contribution to knowledge does it offer.

Title is too long and should be shortened. Gram negative in the title and everywhere in the whole manuscript should be corrected by capitalizing the “G” in “gram”.

Overall, there are some grammar and spelling mistakes which should be carefully revised.

Introduction does not provide enough background information to the topic. There is no referral at all to previous studies performed to investigate the same topic although the topic has been extensively studied. In general, the introduction should be improved.

Line 35: What do authors mean by “typical Gram-negative bacterial pathogens”. The sentence should be re-expressed.

Line 36: What do you mean by “ESBL content”. This should be better expressed. Please use alternative more accurate words.

How did you find 189 Gram negative bacteria from 181 participants? Were there cases of mixed infections? You mention that there were 66 cases of mixed growth. What were your criteria to diagnose mixed infections in these cases? If there were additional organisms found? What were these organisms?

Introduction

Introduction does not provide enough background information to the topic. There is no referral at all to previous studies performed to investigate the same topic although the topic has been extensively studied. In general, the introduction should be improved.

Line 57-58: The definition of RTIs shown is not scientifically accurate. Please refer to medical textbooks or to diagnostic references to include a more accurate definition.

Line 65: “An HAI occurs when a patient is receiving treatment at a medical facility”. Please revise and correct this sentence referring to the definition of HAI and what does it mean?

Line 66-69: The sentence starting with “Among the known types” should be revised and divided into 2 or 3 sentences as the three lines are not understandable.

Line 83: What do you mean by “excessively invasive” ? Please re-word.

Line 85-87: The expression of idea is not clear. Re-write this part.

Methods:

Line 100: What do you mean by “intubated with a mechanical ventilator”?

Line 145-148: What is the rationale for selecting these antibiotics? There are other antibiotics with a spectrum of action on LRTIs which are not included.

How was MDR decided based only on testing these antibiotics? I find these are insufficient to determine MDR. Testing one agent from each class cannot be used to classify MDR.

Line 177-178: “We assessed the relationship between the dependent and independent variables using bivariate analysis. Those with a p-value ≤ 0.25 were candidates for multivariate analysis.” This is not clear. Please identify what do you mean by “bivariate analysis”. What is the reason for choosing this cutoff value?

Results:

Line 198: What do you mean by “developed MDR”? Please re-word.

Line 199: The sentence is not clear. “Overall, 86.8% of the ESBL-producing isolates were produced” Please re-word.

Line 214: “we tested for any statistically significant associations among the parameters.” This sentence is not clear. Which associations are tested? What were the variables studied? Please make the sentence clearer.

Line 218-219: The detailed methodology of bivariate and multivariate logistic regression analyses used must be included in the method section and explained in detail.

Table 1: What do you mean by “Pedi Emergency”? Is it Paediatrics emergency?

Table 2 title: Please change the title into “Identification of isolated bacteria” rather than “bacterial profile”.

Table 4: What does the symbol “I” refer to in the table?

Discussion

Line224-225: Currently, the rapid spread of MDR gram-negative bacterial pathogens in ICUs, which is mainly due to the rapid increase in ESBLs”. Please re-write this sentence correctly. MDR is not mainly due to the rapid increase in ESBLs.

Line 233-235: In which geographic regions were these studies performed? Are these all from the same area?

Line 239: Are all of the mentioned factors are considered confounding factors? Some of these factors are not confounding but are risk factors. Please revise this whole paragraph and re-write accurately to convey the correct meaning.

Line 244: In this paragraph, you mention “the prevalence of Gram-negative bacteria from tracheal aspirate” and following that “Furthermore, the prevalence of gram-negative bacteria from LRTI in our study”. I understand from the method section that you only collected tracheal aspirates, so what is the difference shown here? Why do you list Gram-negative bacteria from “tracheal aspirates” and “LRTI” as two different things?

Line 255: “in agreement with previously reported studies” What are these studies? Where were these studies carried out? What was the of isolates in each study?

Line 256: “were the predominant isolates according to similar studies but differed from previous ones”. Please clarify. What were the similar studies and what were the different studies. The sentence is not clearly written.

Line 327-328: Non- availability of antibiotic discs cannot be considered a limitation, but this is a source of defect in the study because this may affect the basic definition of MDR. Additionally, some important agents must be included for soundness of the study. This is considered basic requirement for routine microbiology work rather than a highly advanced technology that requires funding!!

Line 330: “In this study, we found a high magnitude of gram-negative bacteria, with an overall prevalence of 65.2%” What do you mean by finding a high magnitude? What is meant by this prevalence? Prevalence of what? Please re-write the sentence accurately.

Reviewer #2: Dear authors,

The work is interesting and well-described, but it could be improved in some points I detail below.

- All the tables mentioned are missing.

- Please, review acronyms without definitions (e.g., MDR in line 72).

- Bacterial genus should be defined the first time each species is mentioned in the text. Please, review.

- To improve the work, I suggest characterizing the ESBL detected using PCR.

**Do you want your identity to be public for this peer review?** For information about this choice, including consent withdrawal, please see our Privacy Policy

Reviewer #1: **Yes: ** Wedad M. Nageeb

Reviewer #2: No

---

## [Author Response · Author response to Decision Letter 1]

19 Dec 2024

Reviewers’ comments for plos one

Dear reviewers,

Thank you for taking the time to review our manuscript and for providing such detailed and constructive feedback. We truly appreciate your extensive review and the valuable suggestions you have made. Below, we have addressed each of your point’s point-by-point and updated manuscript accordingly.

Point by point response to Reviewer #1

First of all, thank you for considering the comments and suggestion on manuscript. The manuscripts have seen a lot of improvement made and alien the point-by-point response and revised manuscripts. In order to improve more add some comment and suggestions.

Although the topic is important, I find that it lacks novelty because there are lots of studies that have investigated the prevalence of MDR and ESBL in Gram negative bacteria. Authors need to show how is their work different and what is the important contribution to knowledge does it offer.

Response: Thank you for your valuable feedback and for highlighting the importance of addressing novelty in this area of research. While we agree that numerous studies have investigated. However, our study offers distinct contributions: Of these

Geographic Relevance: This study focuses on Ethiopia, where data on MDR and ESBL prevalence in ICU settings is limited. This study will offer critical insights into the local burden and patterns of resistance. Furthermore, ICU Context this research specifically targets ICU patients with a high-risk group with a unique bacterial resistance profile which provides actionable data for improving infection prevention and control strategies in these units.

And from public health perspective the finding will have implications for antibiotic stewardship programs in resource-limited settings, addressing gaps in national and regional guidelines.

1. Title is too long and should be shortened. Gram negative in the title and everywhere in the whole manuscript should be corrected by capitalizing the “G” in “gram”.

Respond: Accepted and correction has been made, however, capitalizing the “G” in “gram” is not changed in the title because the submission format doesn’t support to capitalize except the first letter of the title

2. Overall, there are some grammars and spelling mistakes which should be carefully revised.

Introduction does not provide enough background information to the topic. There is no referral at all to previous studies performed to investigate the same topic although the topic has been extensively studied. In general, the introduction should be improved.

Respond: Thank you, the comment is well taken and revision has been made and grammar and spelling mistake are improved

3. Line 35: What do authors mean by “typical Gram-negative bacterial pathogens”. The sentence should be re-expressed.

Respond: Revision has been made, (line 33)

4. Line 36: What do you mean by “ESBL content”. This should be better expressed. Please use alternative more accurate words.

Respond: Revision has been made, (line 34)

5. How did you find 189 Gram negative bacteria from 181 participants? Were there cases of mixed infections? You mention that there were 66 cases of mixed growth. What were your criteria to diagnose mixed infections in these cases? If there were additional organisms found? What were these organisms?

Respond: Revision has been made, and 66 (55.9%) were identified as double-organisms gram negative bacteria and we do this based on Standard operating procedures of our Microbiology laboratory (line 46)

Introduction

6. Line 57-58: The definition of RTIs shown is not scientifically accurate. Please refer to medical textbooks or to diagnostic references to include a more accurate definition.

Respond: Corrected and a major revision has been made in the introduction part and this part is removed and improved.

7. Line 65: “An HAI occurs when a patient is receiving treatment at a medical facility”. Please revise and correct this sentence referring to the definition of HAI and what does it mean?

Respond: Corrected and a major revision has been made in the introduction part and this part is removed and improved.

8. Line 66-69: The sentence starting with “Among the known types” should be revised and divided into 2 or 3 sentences as the three lines are not understandable.

Respond: correction has been taken (line 58-59)

9. Line 83: What do you mean by “excessively invasive” ? Please re-word.

Respond: Corrected (line 79-82)

10. Line 85-87: The expression of idea is not clear. Re-write this part.

Response: Comment accepted and corrected accordingly (lines 62-65)

Methods:

11. Line 100: What do you mean by “intubated with a mechanical ventilator”?

Respond: Revised and re-written (line 105-106)

12. Line 145-148: What is the rationale for selecting these antibiotics? There are other antibiotics with a spectrum of action on LRTIs which are not included.

How was MDR decided based only on testing these antibiotics? I find these are insufficient to determine MDR. Testing one agent from each class cannot be used to classify MDR.

Respond: All antibiotics for Gram-negative bacteria were not tested due to the lack of antibiotic discs. However, we selected and included all the antibiotics used as the hospital treatment protocol recommendations in the ICU units ( Lines 336-337).

13. Line 177-178: “We assessed the relationship between the dependent and independent variables using bivariate analysis. Those with a p-value ≤ 0.25 were candidates for multivariate analysis.” This is not clear. Please identify what do you mean by “bivariate analysis”. What is the reason for choosing this cutoff value?

Respond: Revised and corrected accordingly (lines 179-182)

14. Line 198: What do you mean by “developed MDR”? Please re-word.

Respond: Corrected line 201-202

15. Line 199: The sentence is not clear. “Overall, 86.8% of the ESBL-producing isolates were produced” Please re-word.

Respond: Corrected (line 202-203)

16. Line 214: “we tested for any statistically significant associations among the parameters.” This sentence is not clear. Which associations are tested? What were the variables studied? Please make the sentence clearer.

Respond: Corrected and comments incorporated (line 217-225)

17.Lines 218-219: The detailed methodology of bivariate and multivariate logistic regression analyses used must be included in the method section and explained in detail.

Respond: Corrected and comments incorporated (line 179-182)

18. Table 1: What do you mean by “Pedi Emergency”? Is it Pediatrics emergency?

Respond: Accepted and updated accordingly

19. Table 2 title: Please change the title into “Identification of isolated bacteria” rather than “bacterial profile”.

Respond: Thank you for your insightful comment and the revision has been done accordingly.

20. Table 4: What does the symbol “I” refer to in the table?

Respond: Change has been made. "I" was replaced with 1* and it is represent as a reference.

Discussion

21. Line224-225: Currently, the rapid spread of MDR gram-negative bacterial pathogens in ICUs, which is mainly due to the rapid increase in ESBLs”. Please re-write this sentence correctly. MDR is not mainly due to the rapid increase in ESBLs.

Respond: Revision has been done (line 229-230)

22. Line 233-235: In which geographic regions were these studies performed? Are these all from the same area?

Respond: Corrected and study sites were incorporated (line 239)

23. Line 239: Are all of the mentioned factors are considered confounding factors? Some of these factors are not confounding but are risk factors. Please revise this whole paragraph and re-write accurately to convey the correct meaning.

Respond: Thank you, Revision has been done (line 245-248)

24. Line 244: In this paragraph, you mention “the prevalence of Gram-negative bacteria from tracheal aspirate” and following that “Furthermore, the prevalence of gram-negative bacteria from LRTI in our study”. I understand from the method section that you only collected tracheal aspirates, so what is the difference shown here? Why do you list Gram-negative bacteria from “tracheal aspirates” and “LRTI” as two different things? Corrected

Respond: Thank you, Revision has been done (line 252-254)

25. Line 255: “in agreement with previously reported studies” What are these studies? Where were these studies carried out? What was the of isolates in each study?

Respond: Thank you. updated based on the comments (lines 260-267)

26. Line 256: “were the predominant isolates according to similar studies but differed from previous ones”. Please clarify. What were the similar studies and what were the different studies. The sentence is not clearly written.

Respond: Thank you. updated based on the comments (lines 263-267)

27. Line 327-328: Non- availability of antibiotic discs cannot be considered a limitation, but this is a source of defect in the study because this may affect the basic definition of MDR. Additionally, some important agents must be included for soundness of the study. This is considered basic requirement for routine microbiology work rather than a highly advanced technology that requires funding!

Respond: Thank you. updated based on the comments (line 336-337)

28. Line 330: “In this study, we found a high magnitude of gram-negative bacteria, with an overall prevalence of 65.2%” What do you mean by finding a high magnitude? What is meant by this prevalence? Prevalence of what? Please re-write the sentence accurately.

Respond: the comment was approved and we revised it (lines 339-340)

Reviewer#2:

29- All the tables mentioned are missing.

Respond: The comments are addressed.

30- Please, review acronyms without definitions (e.g., MDR in line 72).

Respond: Thank you. the comment is accepted and improved

31- To improve the work, I suggest characterizing the ESBL detected using PCR.

Respond: Thank you for your insightful comment and we accept your comment. For the time being due to resource limitation we can’t do the PCR, however, we have a plan to characterize the ESBL-producer isolates in near future.

---

## [Decision Letter · Decision Letter 1]

24 Jan 2025

Dear Dr. Gebreyohannes,

Thank you for submitting your manuscript to PLOS ONE. After careful consideration, we feel that it has merit but does not fully meet PLOS ONE’s publication criteria as it currently stands. Therefore, we invite you to submit a revised version of the manuscript that addresses the points raised during the review process. Please submit your revised manuscript within Mar 10 2025 11:59PM. If you will need more time than this to complete your revisions, please reply to this message or contact the journal office at plosone@plos.org . A rebuttal letter that responds to each point raised by the academic editor and reviewer(s). You should upload this letter as a separate file labeled 'Response to Reviewers'.A marked-up copy of your manuscript that highlights changes made to the original version. You should upload this as a separate file labeled 'Revised Manuscript with Track Changes'.An unmarked version of your revised paper without tracked changes. You should upload this as a separate file labeled 'Manuscript'.

We look forward to receiving your revised manuscript.

Kind regards,

*
**Ali Amanati**
*

**Academic Editor**

PLOS ONE

Journal Requirements:

Additional Editor Comments:

Dear authors

The overall presentation of the manuscript has significantly improved after the ‎amendments, enhancing its readability. Although the authors have ‎satisfactorily addressed the reviewers' concerns, additional improvements are ‎still required based on the comments posted by Reviewer #3. ‎

Reviewers' comments:

Reviewer's Responses to Questions

**Comments to the Author**

Reviewer #3: (No Response)

Reviewer #4: All comments have been addressed

Reviewer #5: All comments have been addressed

Reviewer #6: All comments have been addressed

Reviewer #7: (No Response)

2. Is the manuscript technically sound, and do the data support the conclusions?

Reviewer #3: Yes

Reviewer #4: Yes

Reviewer #5: Partly

Reviewer #6: (No Response)

Reviewer #7: No

3. Has the statistical analysis been performed appropriately and rigorously?

Reviewer #3: Yes

Reviewer #4: Yes

Reviewer #5: Yes

Reviewer #6: (No Response)

Reviewer #7: No

4. Have the authors made all data underlying the findings in their manuscript fully available?

Reviewer #3: Yes

Reviewer #4: Yes

Reviewer #5: Yes

Reviewer #6: (No Response)

Reviewer #7: No

5. Is the manuscript presented in an intelligible fashion and written in standard English?

Reviewer #3: Yes

Reviewer #4: Yes

Reviewer #5: Yes

Reviewer #6: (No Response)

Reviewer #7: Yes

Reviewer #3: 1. Introduction

Contextual Relevance: While the introduction effectively establishes the global significance of antimicrobial resistance, it lacks sufficient contextualization for Ethiopia. Discuss specific challenges or gaps in surveillance and resistance patterns in the region.

Objectives: Explicitly state the research objectives as testable hypotheses or specific aims to guide the reader.

2. Methods

Study Population:

Defi2ne inclusion/exclusion criteria more clearly, including the rationale for excluding tuberculosis patients and those intubated for less than 48 hours.

Specify whether demographic or clinical factors (e.g., comorbidities, prior antibiotic use) were considered.

Sample Handling:

Provide more details on how sample collection was standardized to minimize contamination.

Include information on how samples were transported and processed to ensure timely and accurate results.

Antimicrobial Susceptibility Testing (AST):

Justify the selection of the antibiotics tested, particularly for carbapenems and aminoglycosides, and explain their relevance to local prescribing practices.

Clarify if and how intermediate susceptibility results were categorized in the analysis.

Statistical Analysis:

Describe the handling of missing data and whether sensitivity analyses were performed.

Explain the rationale for including variables in the logistic regression model and how multicollinearity was assessed.

3. Results

Data Presentation:

Provide stratified data to explore variations in MDR/ESBL prevalence by age, sex, or ICU type (e.g., pediatric vs. adult).

Avoid redundancy by streamlining overlapping information in the text, tables, and figures.

Key Findings:

Highlight the clinical significance of the high resistance rates observed for specific antibiotic classes, particularly carbapenems and cephalosporins.

Clarify if there were significant trends or outliers in resistance patterns.

Comprehensive Review of Manuscript PONE-D-24-12990

Title: Magnitude of multidrug-resistant and extended spectrum β-lactamase-producing gram-negative bacteria in patients with lower respiratory tract infection at intensive care units of St. Paul’s Hospital Millennium Medical College, Addis Ababa, Ethiopia

General Assessment

This manuscript addresses an urgent global health issue by investigating the prevalence and antimicrobial resistance patterns of MDR and ESBL-producing gram-negative bacteria in ICU patients with LRTIs. The study provides valuable data for improving infection control and antimicrobial stewardship in resource-limited settings. However, to reach its full potential, the manuscript requires substantial revisions to enhance its clarity, rigor, and contextual relevance.

Strengths

The study is highly relevant to global public health, especially in the context of rising antimicrobial resistance.

The use of both phenotypic antimicrobial susceptibility testing and ESBL confirmation adds rigor to the findings.

The manuscript provides valuable baseline data for Ethiopia, where such data are often scarce.

Areas for Improvement and Recommendations

1. Introduction

Contextual Relevance: While the introduction effectively establishes the global significance of antimicrobial resistance, it lacks sufficient contextualization for Ethiopia. Discuss specific challenges or gaps in surveillance and resistance patterns in the region.

Objectives: Explicitly state the research objectives as testable hypotheses or specific aims to guide the reader.

2. Methods

Study Population:

Define inclusion/exclusion criteria more clearly, including the rationale for excluding tuberculosis patients and those intubated for less than 48 hours.

Specify whether demographic or clinical factors (e.g., comorbidities, prior antibiotic use) were considered.

Sample Handling:

Provide more details on how sample collection was standardized to minimize contamination.

Include information on how samples were transported and processed to ensure timely and accurate results.

Antimicrobial Susceptibility Testing (AST):

Justify the selection of the antibiotics tested, particularly for carbapenems and aminoglycosides, and explain their relevance to local prescribing practices.

Clarify if and how intermediate susceptibility results were categorized in the analysis.

Statistical Analysis:

Describe the handling of missing data and whether sensitivity analyses were performed.

Explain the rationale for including variables in the logistic regression model and how multicollinearity was assessed.

3. Results

Data Presentation:

Provide stratified data to explore variations in MDR/ESBL prevalence by age, sex, or ICU type (e.g., pediatric vs. adult).

Avoid redundancy by streamlining overlapping information in the text, tables, and figures.

Key Findings:

Highlight the clinical significance of the high resistance rates observed for specific antibiotic classes, particularly carbapenems and cephalosporins.

Clarify if there were significant trends or outliers in resistance patterns.

4. Discussion

Interpretation of Findings:

Delve deeper into the potential drivers of high MDR and ESBL prevalence, such as local antibiotic prescribing practices, infection control gaps, or environmental factors.

Compare findings with a broader range of studies, particularly those from similar healthcare settings in sub-Saharan Africa.

Implications:

Discuss the implications of the findings for empirical treatment guidelines, particularly in resource-limited settings where access to advanced antibiotics may be constrained.

Limitations:

Acknowledge the exclusion of gram-positive bacteria and fungal pathogens, and explain how this impacts the study's generalizability.

Discuss potential biases introduced by the cross-sectional design, such as the inability to establish causality.

Reviewer #4: Suggested changes and queries and comments have been addressed adequately by the authors.Grammatical errors have been corrected .

Reviewer #5: Dear autnors,

Thank you for taking time to address our earlier concerns. I have no further comments at this stage

Reviewer #6: (No Response)

Reviewer #7: The manuscript describes the prevalence of MDR organisms obtained in Tracheal cultures in their setting. There is no justification for selecting this specimen or organ system as MDR organisms are also prevalent in other organ systems and specimen types.

Moreover the study in its present form does not offer a meaningful insight into the management of VAP and / or the reasons for the emergence of MDR organisms in this system.

The authors may do well to have more clinical orientation with an attempt to connect the microbiology and the clinical and radiological evidence of pneumonia.

Insight into other organ systems should be important to better define theprevalence and magnitude of the problem of MDR & ESBL producing GNB in hospital practice

**Do you want your identity to be public for this peer review?** For information about this choice, including consent withdrawal, please see our Privacy Policy

Reviewer #3: No

Reviewer #4: **Yes: ** Reba Kanungo

Reviewer #5: No

Reviewer #6: No

Reviewer #7: No

---

## [Author Response · Author response to Decision Letter 2]

15 Mar 2025

Dear reviewer,

Thank you for taking the time to review our manuscript and for providing such detailed and constructive feedback. We truly appreciate your thorough analysis and the valuable suggestions you have made. Below, we have addressed each of your point’s point-by-point and updated the manuscript accordingly.

---

## [Decision Letter · Decision Letter 2]

31 Mar 2025

Dear Dr. Berhe,

Thank you for submitting your manuscript to PLOS ONE. After careful consideration, we feel that it has merit but does not fully meet PLOS ONE’s publication criteria as it currently stands. Therefore, we invite you to submit a revised version of the manuscript that addresses the points raised during the review process. Please submit your revised manuscript by May 15 2025 11:59PM. If you will need more time than this to complete your revisions, please reply to this message or contact the journal office at plosone@plos.org . A rebuttal letter that responds to each point raised by the academic editor and reviewer(s). You should upload this letter as a separate file labeled 'Response to Reviewers'.A marked-up copy of your manuscript that highlights changes made to the original version. You should upload this as a separate file labeled 'Revised Manuscript with Track Changes'.An unmarked version of your revised paper without tracked changes. You should upload this as a separate file labeled 'Manuscript'.

We look forward to receiving your revised manuscript.

Kind regards,

*
**Ali Amanati**
*

**Academic Editor**

PLOS ONE

Journal Requirements:

Additional Editor Comments:

Dear authors,

Minor correction should be considered:

-Lines 173-175: "Intermediate susceptibility values were considered reduced and treated as resistant in the analysis to ensure appropriate reporting and antimicrobial stewardship guidance." should be corrected. Optional Editor suggestion: "Intermediate susceptibility test results were considered as "resistant spp." to ensure appropriate reporting in accordance with antimicrobial stewardship guidance.", this statement need reference.

-Line 384: "Typhoid resistance" is incomplete statement. Typhoid resistance to which antibiotics? this statement also need reference.

Reviewers' comments:

Reviewer's Responses to Questions

**Comments to the Author**

Reviewer #3: All comments have been addressed

2. Is the manuscript technically sound, and do the data support the conclusions?

Reviewer #3: Yes

3. Has the statistical analysis been performed appropriately and rigorously?

Reviewer #3: Yes

4. Have the authors made all data underlying the findings in their manuscript fully available?

Reviewer #3: Yes

5. Is the manuscript presented in an intelligible fashion and written in standard English?

Reviewer #3: Yes

Reviewer #3: I have reviewed the manuscript and the authors provided detailed responses to the questions. I do not have any additional comments or concerns for the authors.

**Do you want your identity to be public for this peer review?** For information about this choice, including consent withdrawal, please see our Privacy Policy

Reviewer #3: No

---

## [Author Response · Author response to Decision Letter 3]

10 Apr 2025

1. Lines 173-175: "Intermediate susceptibility test results were considered as "resistant spp." to ensure appropriate reporting in accordance with antimicrobial stewardship guidance.", this statement needs reference.

Response: Thank you for your insightful comment, as per your suggestion we removed the referenced statement. And the AST results have been reanalyzed. Following this minimal changes in resistance patterns of Acinetobacter spp. and K.pneumonae species in Table 3 and specifically, the high resistance level originally reported for Acinetobacter spp. across all antimicrobial classes—especially cephalosporins—has been updated from 89.6%–96.9% to 86.5%–95.8% (Line 252).

The reason for the previous categorization of intermediate as resistance was due to by considering the patients' critical clinical condition, antimicrobial treatment must precede documented susceptibility results rather than intermediate susceptibility, which only promises optimal clinical outcomes. The trend in our study site is to treat critical patients only with susceptible drugs. Additionally, according to the reference (Centre for Health Protection, Department of Health - Antimicrobial Resistance (AMR) Surveillance in Public Hospitals and Clinics - Hospital Authority AMR Data (2022) the intermediate and resistant isolates were considered or categorized as non-susceptible in the AMR surveillance report.

2. Line 384: "Typhoid resistance" is an incomplete statement. Typhoid resistance to which antibiotics? this statement also needs reference.

Response: Thank you for your insightful comment. The section has been carefully reviewed and revised accordingly. The necessary edits have been incorporated, and a relevant reference has been added. However, the reference was taken from the WHO report on “Addressing the Challenge of Antimicrobial Resistance in Ethiopia. This doesn’t mention the type of antimicrobial agents”. The antimicrobial resistance level in the causative agent of typhoid fever increased from 10% in 2006 to 87% in 2019 and similarly, E. coli strains showed increased resistance to stronger antimicrobial agents. Line 377-379

---

## [Editor Report · Decision Letter 3]

22 Apr 2025

Magnitude of multidrug-resistant and extended-spectrum β-lactamase-producing gram-negative bacteria from tracheal aspirates of intensive care unit patients in Ethiopia.

PONE-D-24-12990R3

Dear Dr. Zenebe Gebreyohannes Berhe,

We’re pleased to inform you that your manuscript has been judged scientifically suitable for publication and will be formally accepted for publication once it meets all outstanding technical requirements.

Kind regards,

*
**Ali Amanati**
*

**Academic Editor**

PLOS ONE

Additional Editor Comments (optional):

I have no further comments to add. I thank the authors for their very detailed ‎‎reply to my comments.‎

---

## [Editor Report · Acceptance letter]

PONE-D-24-12990R3

PLOS ONE

Dear Dr. Berhe,

I'm pleased to inform you that your manuscript has been deemed suitable for publication in PLOS ONE. Congratulations! Your manuscript is now being handed over to our production team.

Kind regards,

on behalf of

Professor Ali Amanati

Academic Editor

PLOS ONE